# Genetic Diversity and Population Structure of Brown Croaker (*Miichthys miiuy*) in Korea and China Inferred from mtDNA Control Region

**DOI:** 10.3390/genes14091692

**Published:** 2023-08-25

**Authors:** Woo-Seok Gwak, Animesh Roy

**Affiliations:** 1Department of Marine Biology and Aquaculture, The Institute of Marine Industry, Gyeongsang National University, Tongyeong 650-160, Republic of Korea; animesh10@bsmrau.edu.bd; 2Department of Fisheries Biology and Aquatic Environment, Bangabandhu Sheikh Mujibur Rahman Agricultural University, Gazipur 1706, Bangladesh

**Keywords:** *Miichthys miiuy*, control region, population structure, genetic diversity

## Abstract

Brown croaker (*Miichthys miiuy*), a species of fish with significant commercial value, is found in the coastal seas of Korea, China, and Japan. The genetic diversity and population structure of a representative sample of brown croaker specimens were assessed based on the control region of their mitochondrial DNA (mtDNA). Samples from a total of 115 individuals were collected from three separate locations, one in China (Lianyungang) and two in Korea (Mokpo and Gyeongnyeolbiyeoldo Island). Analysis of the 436-base-pair mtDNA control region revealed that the haplotype diversity ranged from 0.973 ± 0.025 to 0.988 ± 0.008, while the nucleotide diversity ranged from 0.012 ± 0.006 to 0.017 ± 0.009. The level of genetic diversity, star-shaped haplotype network, significant Fu’s Fs test, and analysis of the mismatch distribution all suggested that this species has experienced population expansion. Fixation index analysis indicated that the population collected at the site in China differed significantly from the two populations obtained in Korea. The findings of this study extend the general understanding of the population structure of *M. miiuy* and can be used to develop strategies for effective resource management.

## 1. Introduction

*M. miiuy* (brown croaker) is a semi-benthic member of the Sciaenidae family that is found throughout the coastal waters of Korea, China, and Japan on muddy to sandy bottoms at a depth of 15–100 m [1,2,3]. In Korea, brown croaker has two main spawning grounds, one located around Deokjeokdo, Incheon, which is the northern part of the West Sea of Korea, and the other located around Jindo Island, in the southwestern waters of Korea. The East China Sea is the primary spawning area in China. In Korea, the spawning season runs from July to September, whereas the spawning season in China has been reported to occur at different times, such as between September and November along the east coast of China, from September to October in Sanduao Bay, and from May to June in Zhejiang [4,5,6,7,8,9]. In Korea, brown croaker has been reported to migrate south during winter and then migrate north again in spring [4]. Brown croaker is a commercially important fish in Korea, China, and the Northwestern Pacific region due to its high nutritional and medicinal value and its flavor [5,10,11]. However, the wild population of brown croaker has drastically declined because of overfishing, and, in many regions in China, fishing grounds have virtually disappeared [12].

The development of suitable management strategies for the sustainable use of resources relies on an understanding of the genetic structure of fish populations, investigations of which also advance the general understanding of biotic evolution [13,14]. Although knowledge of the population structure is crucial for management and conservation, the brown croaker remains relatively under-studied. Xu et al. [15] conducted an experiment in China using the cytochrome oxidase subunit I gene (COI) and found insignificant genetic differentiation among brown croaker populations. Conversely, Cheng et al. [16] conducted research along the coast of the East China Sea in which specimens were taken from six locations (Ruian, Wenling, Wenzhou, Xiangshan, Yueqing, and Zhoushan), and their mitochondrial DNA (mtDNA) control regions were compared. Their work revealed no structural distinctions based on *F*_ST_ analysis, with the exception of one population (Zhoushan) despite the fact that no obvious physical barriers existed between the sampled populations. However, to the best of our knowledge, no studies of the brown croaker population structure have been conducted in Korean waters. Therefore, this study represents an attempt to characterize the population structure in Korean waters and to discern whether these populations differ significantly from those in China.

Many molecular marker techniques have been developed in recent years to investigate the genetic structure of plant and animal populations for conservation and/or to ensure the sustainable yields of commercial species [17]. In particular, mtDNA has frequently been used as a sensitive marker for population structure analysis due to its maternal inheritance, the absence of recombination, and a faster evolutionary rate than nuclear DNA [18]. The control region of mtDNA is a more potent and reliable determinant of genetic diversity and population structure in marine fish species than other conventional mtDNA markers [19,20,21]. Therefore, the mtDNA control region was used as a marker in the present study. Samples were collected from two locations in Korea and one location in China in order to investigate the genetic diversity and population structure of brown croaker within these regions.

## 2. Materials and Methods

### 2.1. Sample Collection

A total of 115 individual wild brown croakers (Figure 1) were collected from Mokpo (MMM) and Gyeongnyeolbiyeoldo (MMB) in Korea and from Lianyungang (MMC) in China between 2008 and 2016 (Figure 2; Table 1). All individual specimens were preserved at −80 °C. Muscle samples were collected from each individual and stored in 99% ethanol until DNA extraction.

### 2.2. DNA Extraction and mtDNA Analysis

Genomic DNA was extracted using proteinase K and a Wizard genomic DNA purification kit (Promega, Madison, WI, USA). A pair of universal primers, L15926 (5′-TCAAAGCTTACACCAGTCTTGTAAACC-3′) [22] and H-16498 (5′-CCTGAAGTAGGAACCAGATG-3′) [23], were used to amplify the mtDNA control region. A 15 µL volume of the reaction mixture containing template DNA (0.6 µL), 10 Ex Taq DNA polymerase buffer (1.5 µL; Takara, Otsu, Japan), 5 µM primers (1.5 µL for each forward and reverse primer), 2.5 mM deoxyribonucleoside triphosphate (dNTP; 1.5 µL), and Ex Taq DNA polymerase of 0.1 µL (Takara) was used for the polymerase chain reaction (PCR). Sterilized water was used to make up the remainder of the final 15 µL volume. The following PCR amplification conditions were used: initial denaturation for 5 min at 94 °C, followed by 30 cycles with denaturation for 30 s at 94 °C, annealing for 30 s at 50 °C, extension for 30 s at 72 °C, and final extension for 7 min at 72 °C. The PCR products were verified against a standard-size marker via gel electrophoresis using 1.5% agarose gel. Further purification was performed on the PCR products to remove extra dNTPs and single-stranded primer residue. Using an ABI BigDye Terminator Cycle Sequencing Ready Reaction Kit v3.1 (Applied Biosystems Inc., Foster City, CA, USA), both of the strands were then sequenced on an ABI 3730XL DNA Analyzer (Applied Biosystems Inc., Foster City, CA, USA).

### 2.3. Data Analysis

The 115 brown croaker mtDNA control region sequences were checked and aligned using clustalW [24], which was implemented in the Bioedit software package (version 7.2.3) [25]. The computer application ARLEQUIN (version 3.5) was used to calculate genetic diversity indices, including the number of polymorphic sites, haplotype diversity, and nucleotide diversity [26]. Pairwise genetic variation and the population structure were assessed using the fixation index (*F*_ST_), and its significance was tested using 1000 bootstraps. To observe the relationship among haplotypes, a phylogenetic network was constructed using the obtained haplotypes with the neighbor-joining method [27] implemented in MEGA 5 [28]. Complete deletion was implemented as an option for gaps in MEGA 5. The genetic distances between haplotype sequences for phylogenetic reconstruction were generated using Kimura’s two-parameter model [29] in MEGA 5. The robustness of the phylogenetic relationship determined using the neighbor-joining method was evaluated with 1000 bootstrap replicates. Bootstrap support values above 50 for the major branches are presented in the final tree. As an outgroup, the *Bahaba taipingensis* control region of the mtDNA sequence was employed. To observe the phylogenetic relationship, a maximum-likelihood-based phylogenetic tree was also constructed in MEGA 5. Geographical relationships between the haplotypes were also investigated by constructing a minimum-spanning network using the ARLEQUIN and Hapstar software programs [30]. Genetic signatures for the population demographics of brown croaker were investigated using Tajima’s *D* test [31] and Fu’s *F*s test [32] for selective neutrality in ARLEQUIN. Significantly negative Tajima’s *D* and Fu’s *F*_s_ values were interpreted as signatures of demographic expansion, which was further investigated using mismatch distribution analysis. The distribution is unimodal for a population that has undergone population expansion, whereas it is multimodal for a population at demographic equilibrium. Neutrality tests and mismatch distributions were calculated in AELEQUIN.

The demographic expansion parameter τ (time since expansion expressed in units of mutational time) [33] was calculated to determine the time since expansion using the formula τ = 2 *ut*, where *u* is the mutation rate for the whole sequence under study per generation and *t* is the time measured in generations since expansion. For marine fishes, the molecular clock has not been precisely determined for the control region and varies between different taxonomic groups. The control region of mtDNA is assumed to mutate at the same rate as the protein-coding region, with mutation rates observed for different fish species such as snooks (3.6% divergence per million years (MY)) [34], East African cichlids (2.2–4.5%/MY) [35], and Australian rainbowfish (3%/MY) [36]. However, the control region appears to mutate much faster in some bony fishes, including Lake Malawi cichlids (6.5–8.8%/MY) [37] and Arctic charr (5–10%/MY) [38]. Considering its generation time and body size, it can be assumed that *M. miiuy* has a faster molecular clock as observed for other vertebrates [39]. In the present study, a divergence rate of 5–10%/MY for the mtDNA control region was used to determine the time since population expansion.

## 3. Results

### 3.1. Genetic Diversity

After checking and aligning the 115 sequences, a 436 bp segment of the mtDNA control region was obtained. A total of 87 haplotypes were detected with 79 substitutions. The haplotypes were deposited in the NCBI under the accession number OR142674-OR142760. The haplotype diversity ranged from a low of 0.973 ± 0.025 in Lianyungang to a high of 0.988 ± 0.008 in Mokpo, while the nucleotide diversity ranged from a low of 0.012 ± 0.006 in Mokpo to a high of 0.017 ± 0.009 in Lianyungang. The pooled haplotype and nucleotide diversity were 0.987 ± 0.004 and 0.015 ± 0.008, respectively (Table 1). The nucleotide proportions (A, T, C, and G) were 36.80%, 28.90%, 22.07%, and 12.23%, respectively. Of the 87 haplotypes, five (MMM01, MMM15, MMM18, MMM22, and MMM24) were shared by more than one population. One of these five haplotypes, MMM24 was shared by all populations. In contrast, 82 haplotypes were found in only one population, of which 33, 34, and 15 were found in Mokpo, Gyeongnyeolbiyeoldo, and Lianyungang, respectively. The haplotype distribution among the populations is presented in Table 2.

### 3.2. Demographic History

The minimum spanning tree had a starlike topology, with several haplotypes radiating from common haplotypes, indicating that brown croaker has undergone population expansion. One haplotype was shared by the three populations, which may represent the central haplotype (Figure 3). The neighbor-joining-based (Figure 4a) and maximum-likelihood-based (Figure 4b) phylogenetic trees were constructed from 87 haplotypes, with no significant genealogical structural differences between the sampling locations. The mismatch distribution of the Mokpo population was unimodal, whereas that of the Gyeongnyeolbiyeoldo and Lianyungang populations and of the pooled sample was slightly bimodal (Figure 5). However, significantly negative Fu’s *F*s tests were indicative of population expansion, which was also supported by a low and insignificant Harpending’s raggedness index and sum-of-square deviation (Table 3). The *M. miiuy’s* population expansion was calculated to have occurred between 65,000 and 125,000 years before the present based on the τ value.

### 3.3. Population Structure

The fixation index *F*_ST_ was used to genetically differentiate the sampled populations. The Lianyungang population was found to be significantly different from the Mokpo and Gyeongnyeolbiyeoldo populations (Table 4).

## 4. Discussion

The genetic diversity of a specific fish species is important for its potential adaptation to future conditions and the overall fitness of its individual populations [40,41]. In the present study, we investigated the genetic diversity of brown croaker sampled from three different sites in Korea and China. These brown croaker populations were found to exhibit a high level of haplotype diversity (0.973 ± 0.025 to 0.988 ± 0.008) and a moderate level of nucleotide diversity (0.012 ± 0.006 to 0.017 ± 0.009). Similar patterns of genetic diversity have also been reported for other marine fish species, such as *Microstomus pacificus*, *Tanakius kitaharai*, and *Cleisthenes herzensteini* [42,43,44]. Many factors can potentially contribute to a high haplotype diversity, such as a large population size, environmental variation, and life history characteristics that lead to rapid population growth. Brown croaker is widely distributed in the Northwestern Pacific and is frequently observed in the shallow waters of Korea, China, and Japan, [1,2] indicating that its high population size may be responsible for the high haplotype diversity reported in the present study [18].

The demographic history of brown croaker in the Northwest Pacific is largely unknown. The phylogenetic tree constructed using the neighbor-joining method was shallow, with short branches and low bootstrap support. The haplotypes from individuals at different locations were scattered throughout the tree, which indicates that the species has undergone recent population expansion, which was supported by the star-shaped minimum-spanning tree. However, the mismatch distribution for the Mokpo population was unimodal, whereas the Gyeongnyeolbiyeoldo and Lianyungang populations had a slightly bimodal mismatch distribution. The significantly negative results for the Tajima’s *D* (Lianyungang) and Fu’s *F*s tests and the low and insignificant Harpending’s raggedness index and sum-of-square deviation were all consistent with population expansion [44]. The high haplotype diversity with moderate-to-low nucleotide diversity observed in the present study was also suggestive of population expansion. The population expansion of this species was estimated in the present study to have occurred during the Late Pleistocene period, which was characterized by a series of large glacial–interglacial cycles [45,46]. During this period, changes in the temperature and salinity, global changes in the ocean circulation patterns, and associated changes in productivity are likely to have had an impact on the demographic history of brown croaker. Most of the areas of the East China Sea, Yellow Sea, and coastal regions of the East Sea were exposed during the Late Pleistocene period. Consequently, the prevalence and distribution of brown croaker were presumably heavily impacted by range contraction and expansion. During the Pleistocene Glacial period, brown croaker may have survived in refugia and then expanded to their previous locations when environmental conditions were favorable again. Population expansion was also observed for brown croaker in a previous study [16] in the East China Sea that examined the mtDNA control region of marine fish species in the Northwest Pacific region, including *Larimichthys polyactis* [47], *Hexagrammos otakii* [48], and *Clupea pallasii* [49].

Marine fish species usually exhibit high levels of gene flow and shallow population structures due to the absence of obvious physical barriers to genetic exchange, their high dispersal capabilities, and large effective population sizes [50]. However, the results of the present study were not in accordance with this general characterization of marine fish populations. Population structure analysis indicated that the population from China was significantly different from the two Korean populations. On the other hand, based on the minimum spanning tree and the haplotype distribution, excluding rare haplotypes, there were three haplotypes (MMC04, MMC13, and MMC20) that were shared only by individuals within the Chinese population. Two haplotypes were shared by two individuals and another was shared by three individuals, indicating a reduced gene flow. In contrast, two larger haplotypes (MMM01 and MMM15) were shared only by the two populations in Korea, indicating restrictions in the mixing of the populations. In the present study, we observed genetic differentiation in the *F*_ST_ analysis but no strong phylogenetic break, which may be due to recent population expansion. Only one haplotype (MMM24) was shared by all three populations, though the frequency of occurrence within the Chinese population was very low. The heterogenic frequency of this haplotype suggests a non-panmictic population [51,52]. Many factors may contribute to population differentiation in marine environments, including breeding behavior, water current patterns, and water temperature [53,54]. A previous study [16] of brown croaker that employed the mtDNA control region for population comparisons found no significant differences between all but one population according to *F*_ST_ values. Xu et al. [15] conducted an experiment with cytochrome oxidase subunit I gene (COI) including the Yellow Sea and China Sea and identified panmictic populations.

In the present study, the study locations were in China and Korea, thus covering a broad geographical range for the first time for this species. Similar results over a broad geographical range have been observed for *Synechogobius ommaturus* between Chinese and Korean populations [52] and *C. pallasii* between Chinese and Japanese populations [51]. Brown croakers in China usually spawn at different times in different places such as September to November, September to October, and May to June on the east coast of the East China Sea, Sanduao Bay, and Zhejiang, respectively, [6], whereas spawning season in Korea typically takes place between July and September [4,5,7]. Therefore, the brown croaker populations of Korea and China may be differentiated due to adaptation to different environments. Brown croaker inhabits shallow muddy and sandy bottoms and spawns in shallow coastal waters [16,55]. Previous studies have suggested that fish species inhabiting coastal zones are characterized by dispersal strategies that minimize the movement of larvae to offshore areas [52] and *M. miiuy* may follow these to maximize their retention, survival, and self-recruitment. In marine fish, their early life history influences their survival [56] and plays a crucial role in population differentiation [57].

Lianyungang is a pocket-shaped area that is influenced by coastal currents. The China coasts are strongly influenced by the Yellow Sea Coastal Current and the China Sea Coastal Current [58], and some local currents form gyres. The Subei Coastal Current, a local coastal current running to the north in the southern Yellow Sea in the opposite direction to the Yellow Sea Coastal Current, generates a mesoscale anticyclonic eddy offshore of Qingdao-Shidao [59,60,61]. This local gyre may lead to the retention of larvae and genetic differentiation between the populations. Ocean current patterns have produced structured populations in fish and other species such as *H. otakii* [48], *Sebastes miniatus* [62], *Gadus macrocephalus* [63], and *Charybdis japonica* [60]. The west coast of Korea is strongly influenced by the Yellow Sea Warm Current, which may cause mixing among the two brown croaker populations of Korea (Mokpo and Gyeongnyeolbiyeoldo). Additionally, these two populations may use the same area as a breeding ground and may not migrate long distances; thus, if they are restricted only to the spawning and overwintering grounds, this could encourage mixing between the populations, as has been observed for *G. macrocephalus* [63].

However, mtDNA contains a low number of loci, which can make it difficult to accurately distinguish between populations. Because microsatellite DNA (msDNA) is highly polymorphic, it is often used as a sensitive marker in fish population studies. Therefore, it will be necessary to conduct additional studies on the population structure of brown croaker using sensitive markers such as msDNA for samples collected from various regions in Korea and China. In addition, the number of individuals collected from China was lower than for the two locations in Korea. Therefore, the collection of larger samples from a broader range of Chinese waters may gain clearer insights into the population structure of *M. miiuy*.

## Figures and Tables

**Figure 1 genes-14-01692-f001:**
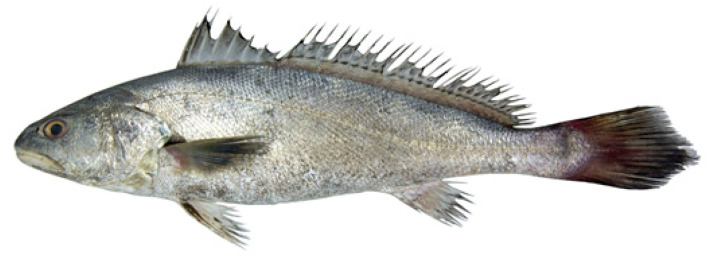
*M. miiuy*.

**Figure 2 genes-14-01692-f002:**
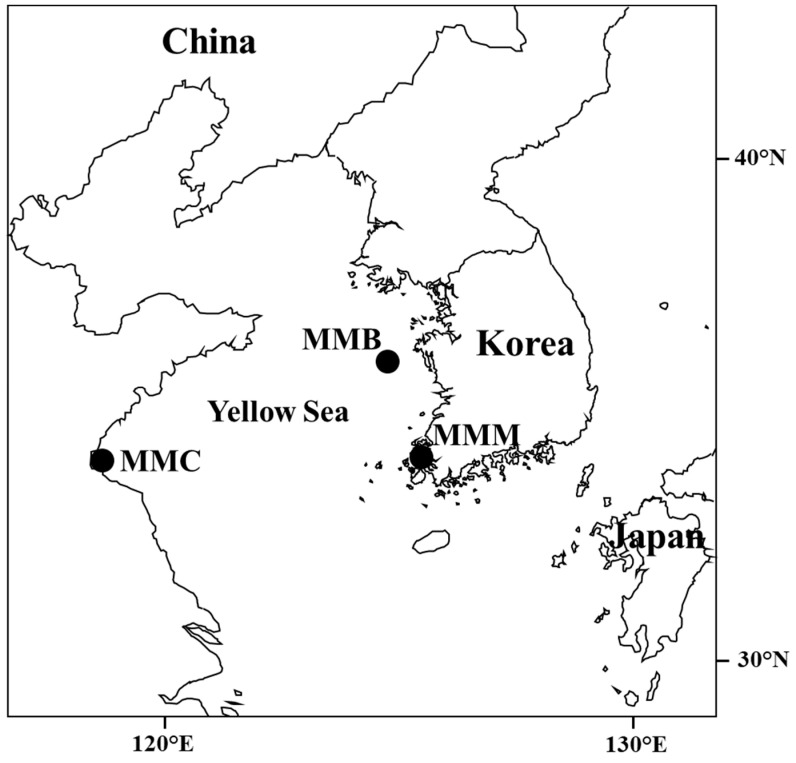
Map showing sample locations for *M. miiuy* for this study. MMM: Mokpo; MMB: Gyeongnyeolbiyeoldo; MMC: Lianyungang.

**Figure 3 genes-14-01692-f003:**
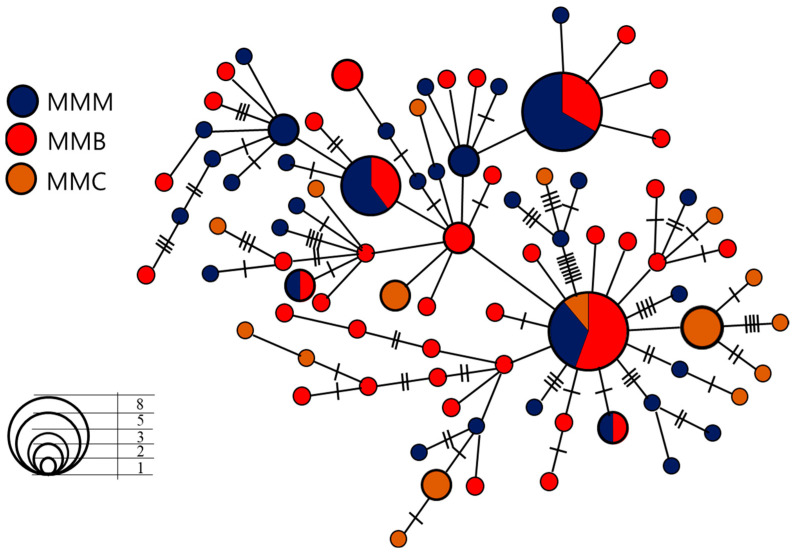
Minimum spanning network based on the control region of mtDNA haplotypes of *M. miiuy*. Circle sizes represent the frequencies of each haplotype. The different colors indicate different populations. Each line connecting haplotypes represents one mutational step.

**Figure 4 genes-14-01692-f004:**
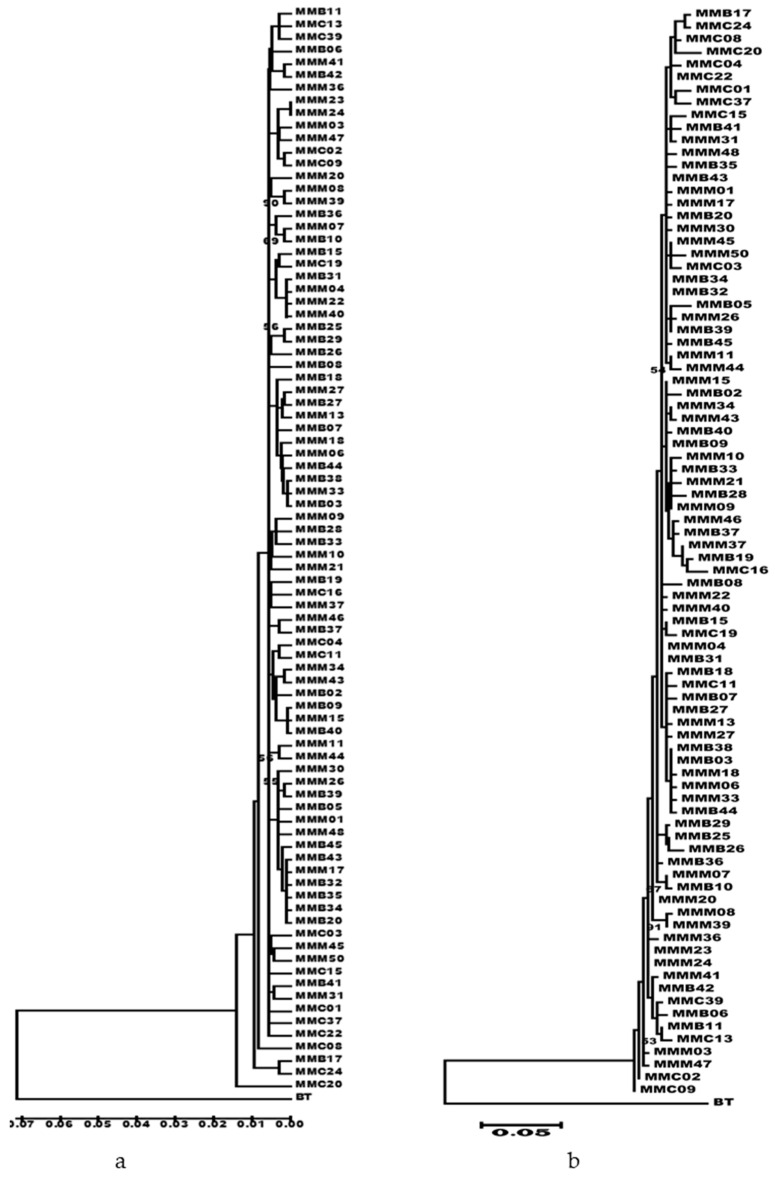
Phylogenetic tree of *M. miiuy* based on the mtDNA control region haplotypes. (**a**) Neighbor-joining-based; (**b**) Maximum-likelihood-based phylogenetic tree. MMM: Mokpo, MMB: Gyeongnyeolbiyeoldo, MMC: Lianyungang. Bootstrap support values above 50 for the different nodes are shown. Bootstrap percentage values are indicated next to nodes of 1000 bootstrap simulations. BT: *B. taipingensis* was used as an outgroup.

**Figure 5 genes-14-01692-f005:**
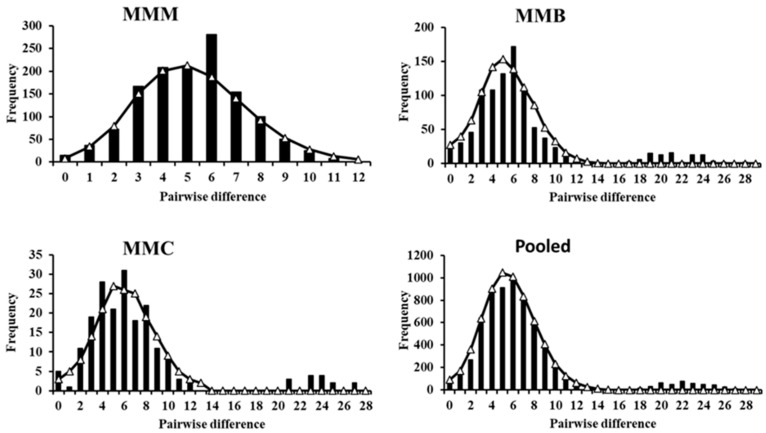
Mismatch distribution of the *M. miiuy* for three populations and pooled sample under the sudden expansion model based on mtDNA control region sequence. Solid lines and bars show the expected distribution and the observed distribution, respectively.

**Table 1 genes-14-01692-t001:** Sampling locations, dates, individual numbers, and molecular diversity indices of *M. miiuy*.

Country	Sampling Sites (Abbreviation)	Collection Date	*N*	*N* _h_	S	*h* (SD)	*π* (SD)
Korea	Mokpo (MMM)	2016, Aug., Sep., Oct.	50	38	45	0.988 ± 0.008	0.012 ± 0.006
	Gyeongnyeolbiyeoldo (MMB)	2017, Jun.	45	39	46	0.976 ± 0.013	0.016 ± 0.008
China	Lianyungang (MMC)	2008, Nov. 7	20	16	36	0.973 ± 0.025	0.017 ± 0.009
Pooled			115	87	79	0.987 ± 0.004	0.015 ± 0.008

Abbreviation: *N*—number of individual samples; *N*_h_—number of haplotypes; S—number of substitutions; *h*—haplotype diversity; *π*—nucleotide diversity; SD—standard deviation.

**Table 2 genes-14-01692-t002:** Haplotype distribution table for the mtDNA control region sequences of *M. miiuy*.

**Location**	**Haplotypes**												
	MMM01	MMM03	MMM04	MMM06	MMM07	MMM08	MMM09	MMM10	MMM11	MMM12	MMM13	MMM15	MMM17
MMM	3	1	2	1	1	1	1	1	1	1	2	6	1
MMB	2	0	0	0	0	0	0	0	0	0	0	2	0
MMC	0	0	0	0	0	0	0	0	0	0	0	0	0
Total	5	1	2	1	1	1	1	1	1	1	2	8	1
	MMM18	MMM19	MMM20	MMM21	MMM22	MMM23	MMM24	MMM26	MMM27	MMM30	MMM31	MMM33	MMM34
MMM	1	1	1	2	1	1	3	1	1	1	1	1	1
MMB	1	0	0	0	1	0	4	0	0	0	0	0	0
MMC	0	0	0	0	0	0	1	0	0	0	0	0	0
Total	2	1	1	2	2	1	8	1	1	1	1	1	1
	MMM36	MMM37	MMM39	MMM40	MMM41	MMM43	MMM44	MMM45	MMM46	MMM47	MMM48	MMM50	MMB02
MMM	2	1	1	1	1	1	1	1	1	1	1	1	0
MMB	0	0	0	0	0	0	0	0	0	0	0	0	1
MMC	0	0	0	0	0	0	0	0	0	0	0	0	0
Total	2	1	1	1	1	1	1	1	1	1	1	1	1
	MMB03	MMB05	MMB06	MMB07	MMB08	MMB09	MMB10	MMB11	MMB15	MMB17	MMB18	MMB19	MMB20
MMM	0	0	0	0	0	0	0	0	0	0	0	0	0
MMB	1	1	1	1	1	1	1	2	1	1	1	1	1
MMC	0	0	0	0	0	0	0	0	0	0	0	0	0
Total	1	1	1	1	1	1	1	2	1	1	1	1	1
	MMB25	MMB26	MMB27	MMB28	MMB29	MMB31	MMB32	MMB33	MMB34	MMB35	MMB36	MMB37	MMB38
MMM	0	0	1	0	0	0	0	0	0	0	0	0	0
MMB	1	1	1	1	1	1	1	1	1	1	1	1	1
MMC	0	0	0	0	0	0	0	0	0	0	0	0	0
Total	1	1	1	1	1	1	1	1	1	1	1	1	1
	MMB39	MMB40	MMB41	MMB42	MMB43	MMB44	MMB45	MMC01	MMC02	MMC03	MMC04	MMC08	MMC09
MMM	0	0	0	0	0	0	0	0	0	0	0	0	0
MMB	1	1	1	1	1	1	1	0	0	0	0	0	0
MMC	0	0	0	0	0	0	0	1	1	1	3	1	1
Total	1	1	1	1	1	1	1	1	1	1	3	1	1
	MMC11	MMC13	MMC15	MMC16	MMC19	MMC20	MMC24	MMC37	MMC39				
MMM	0	0	0	0	0	0	0	0	0				
MMB	0	0	0	0	0	0	0	0	0				
MMC	1	2	1	1	1	2	1	1	1				
Total	1	2	1	1	1	2	1	1	1				

**Table 3 genes-14-01692-t003:** Neutrality test and mismatch distribution for *M. miiuy*.

	Neutrality Test		Mismatch Distribution		
Location	Tajima’s *D*	Fu’s *F*s	SSD	*Hri*	*τ*
MMM	−1.853 *	−25.499 *	0.001	0.013	5.277
MMB	−1.994 *	−22.012 *	0.006	0.015	5.804
MMC	−1.524	−5.487 *	0.008	0.029	5.619
Pooled	−2.016 *	−24.935 *	0.001	0.009	5.468

Abbreviation: SSD—the sum of squared deviations; *Hri*—Harpending raggedness index; *τ*—the time since expansion expressed in units of mutational time; *—Starmarks indicate a significant value (*p* < 0.05).

**Table 4 genes-14-01692-t004:** Pairwise *F*_ST_ (below diagonal) and associated *p* values (above diagonal) among three sample locations for the mtDNA control region of *M. miiuy*.

Location	MMM	MMB	MMC
MMM		0.086	0.000
MMB	0.010		0.000
MMC	**0.163 ***	**0.142 ***	

* Bold letters indicate significance level (*p* < 0.05).

## Data Availability

Available upon request.

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
