# Peer review of "Genetic Diversity and Population Structure of Brown Croaker (Miichthys miiuy) in Korea and China Inferred from mtDNA Control Region"

_genes, 2023, doi:10.3390/genes14091692_

Round 1

Reviewer 1 Report

General Comments

This study presents results from a population-level sequencing survey of the mitochondrial control region for three populations of brown croaker in China and Korea. It is unclear how these results differ from previous mtDNA surveys of brown croaker.  The introduction states that an earlier study investigated mtDNA COI in six populations,but found little structure. It is not clear how results from the present study differ from earlier findings. The addition of mitochondrial loci is not necessarily warranted as all mitochondrial genes are inherited as a single molecule.  Nuclear loci (SNPs or microsatellites) are needed to make more robust conclusions.

Specific Comments

1.      I recommend adding a picture of the study organism as many readers may not be familiar with this species.

2.      Provide a clearer explanation here. How different was this population and where was it located?

Line 50. This work revealed no structural distinctions, with the exception of one population, in spite of the fact that no obvious physical barriers existed between the populations sampled.

3.      Line 100 – A reference is needed for ClustalW alignment.

4.      What is edition?

Line 121 – “After edition and alignment of 115 sequences, a 436 bp segment of the mtDNA control region was obtained.”

5.      How was this divergence rate calibrated?  This requires justification and references.

Line 117 - In the present study, a divergence rate of 5–10%/MY for the 117 mtDNA control region was used to determine the time of population expansion.

6.      Line 144 - I could not find an explanation of the variable τ.  Is this just the percent divergence? 

7.      Line 87 – I do not see any details in the methods regarding the phylogenetic analysis.  Why neighbor joining?  There are no support values at nodes.  It would be helpful of location names could be added as annotations. A model-based approach should also be used (Bayesian or maximum likelihood).

8.      Line 185 – What is expected in a phylogeny when a population has experienced expansion?  Explain.

9.      Line 211 – This states that significant geographic structure was found in the dataset. What is the conclusion based on? Haplotypes were not population specific.  Pairwise FST, while statistically, significant was still low. 

Some phrases are confusing or poorly worded.

Reviewer 2 Report

The authors compared the genetic variation of Miichthys miiuy between Korea and China using mtDNA control region segments.  The details are as follows:

1.     Table 1 should remove to the section 3.1.

2.     The sampling time span is too large, and the individual numbers is also uneven. Especially only 20 individuals were collected from China in 2008. Therefore, I think the research value based on the above sample information were insufficient.

3.     The primer information of L-15924 was incorrect. The primer in references [20] was L15926 (5'-TCAAAGCTTACACCAGTCTTGTAAACC-3'). Please check it.

4.     The authors mentioned that the divergence rate was 5–10%/MY for the mtDNA CR. Please supply the according references. Besides, the calculation formula for the population expansion time should also be provided.

5.     In discussion, L212-213, “Population structure analysis indicated that the population from China was significantly different from the two Korean populations. This result was largely related to sampling time and sample size of Lianyungang.

6.     The accuracy of reference [42] needs to be verified. It is a Chinese periodical, and the paper was not written in Korean.

7.     L221, inappropriate sentence expression. The breeding time of Miichthys miiuy varies in different sea areas of China. The spawning time is from April to June in the Southern East China Sea, but from July to August in the sea area off the Yangtze River Estuary. In the northern Yellow Sea and Bohai Sea, the spawning time is from September to October.

8.     L226-231, There are many errors in the explanation of ocean currents.

 The quality of English language is moderate.

Round 2

Reviewer 2 Report

The authors have made in-depth revisions to the article and the questions proposed by the reviewers are almost answered one by one. The quality of this paper has been greatly improved through revisions. But there are still some minor problems that need to be noted.

1. Table 2 is not necessary, because it has the same meaning with Fig.3. Besides,it took up too many layouts.

2. Please merge Fig. 4 and Fig.5.

Correct language description was used in this paper.
